# Approaching future rewards or waiting for them to arrive: Spatial representations of time and intertemporal choice

Daniel Fletcher [1]*, Robert Houghton[2], Alexa Spence[1]

**1** Department of Psychology, University of Nottingham, Nottingham, United Kingdom, **2** Human Factors Research Group, Faculty of Engineering, University of Nottingham, Nottingham, United Kingdom

* lpxdf2@nottingham.ac.uk

**Data Availability Statement:** All data underlying the results are available from UK Data service ReShare (https://reshare.ukdataservice.ac.uk/856861/).

## Abstract

Our mental representation of the passage of time is structured by concepts of spatial motion, including an ego-moving perspective in which the self is perceived as approaching future events and a time-moving perspective in which future events are perceived as approaching the self. While previous research has found that processing spatial information in one's environment can preferentially activate either an ego-moving or time-moving temporal perspective, potential downstream impacts on everyday decision-making have received less empirical attention. Based on the idea people may feel closer to positive events they see themselves as actively approaching rather than passively waiting for, in this pre-registered study we tested the hypothesis that spatial primes corresponding to an ego-moving (vs. time-moving) perspective would attenuate temporal discounting by making future rewards feel more proximal. 599 participants were randomly assigned to one of three spatial prime conditions (ego-moving, time-moving, control) resembling map-based tasks people may engage with on digital devices, before completing measures of temporal perspective, perceived wait time, perceived control over time, and temporal discounting. Partly consistent with previous research, the results indicated that the time-moving prime successfully activated the intended temporal perspective–though the ego-moving prime did not. Contrary to our primary hypotheses, the spatial primes had no effect on either perceived wait time or temporal discounting. Processing spatial information in a map-based task therefore appears to influence how people conceptualise the passage of time, but there was no evidence for downstream effects on intertemporal preferences. Additionally, exploratory analysis indicated that greater perceived control over time was associated with lower temporal discounting, mediated by a reduction in perceived wait time, suggesting a possible area for future research into individual differences and interventions in intertemporal decision-making.

## Introduction

Spatial metaphors for time are ubiquitous in the English language [1]. We look *forward* to weekends, deadlines *approach*, and days may feel *long* or *short*. In addition to extensive use of

**Funding:** This research was supported by Economic and Social Research Council [grant number ES/P000711/1]. The funders had no role in study design, data collection and analysis, decision to publish, or preparation of the manuscript.

**Fig 1.** Representation of ego-moving (top) and time-moving (bottom) temporal perspectives, with an example of an associated linguistic metaphor.

such linguistic metaphors, research has found that processing spatial information influences how people subsequently interpret temporal information, suggesting an overlap in how these concepts are represented in human cognition [2, 3]. Our mental representation of the passage of time is thought to be structured by concepts of spatial motion, including an ego-moving temporal perspective in which the self is perceived as approaching future events (e.g., "we're approaching the weekend"), and a time-moving temporal perspective in which future events are perceived as approaching the self (e.g., "the weekend is approaching") [1–20] (see Fig 1).

In a seminal paper investigating the relationship between temporal and spatial concepts, Boroditsky [5, Study 1] presented one group of participants with a series of diagrams depicting a person moving between two static objects (ego-moving primes), and another group of participants with a series of diagrams depicting two objects in motion—with one following the other (time-moving primes). Participants were then asked which day of the week a meeting originally scheduled for next Wednesday would take place now it had been moved forward two days. This question, developed by McGlone and Harding [6], is ambiguous because moving an event forward could mean either making it earlier or later, depending on whether forward movement is conceptualised as being towards the future (as in the ego-moving perspective) or towards the present (as in the time-moving perspective) (see Fig 1). Monday responses are therefore assumed to reflect a time-moving perspective, while Friday responses are assumed to reflect an ego-moving perspective [6]. If people draw upon spatial concepts to mentally represent time, the spatial primes in Boroditsky's study should have influenced how participants responded to McGlone and Harding's Monday-Friday question. This was indeed the case, with 73% of participants who saw ego-moving primes indicating the meeting would take place on Friday, compared to 31% of participants who saw time-moving primes, and 54% of participants in an unprimed control condition (see [7] for a direct replication).

## Temporal perspective and perceived control

If people can conceptualise the passage of time in different ways, it is possible that future events feel subjectively closer or further away depending on the temporal perspective adopted [8]. In particular, ego-moving and time-moving perspectives may influence perceived temporal distance by eliciting differing levels of perceived control, with greater perceived control associated

with mental representations of the self actively approaching future events (ego-moving) rather than passively waiting for them to arrive (time-moving) [4]. Such a relationship between perceived control and temporal perspective could emerge from associations developed in the spatial domain, where people are likely to feel greater control over their distance to objects they are approaching compared to objects that are approaching them. Activating spatial concepts relating to self-movement may therefore promote a sense of control over distance, which overgeneralises to the temporal domain even though control over objective temporal distance is not possible.

Consistent with the idea that ego-moving and time-moving perspectives are associated with differing levels of perceived control, Richmond et al. [9] found that participants with a greater sense of personal agency were more likely to provide ego-moving responses to McGlone and Harding's [6] Monday-Friday question. Furthermore, Loermans et al. [10] found that participants who wrote about past events where they had experienced high (vs. low) control were subsequently more likely to adopt an ego-moving perspective (also see [11]). However, while priming higher perceived control therefore appears to activate an ego-moving perspective [10, 11], this does not necessarily mean the reverse relationship exists (i.e., that activating an ego-moving perspective increases perceived control), which has yet to be empirically tested [10].

## Perceived control and perceived temporal distance

While aversive future events typically feel more temporally proximal than positive future events [21], Han and Gershoff [22] found that this tendency was reversed when high (vs. low) perceived control was experimentally induced, with positively valenced events subsequently being perceived as more proximal than negatively valenced events. Based on Han and Gershoff's [22] findings, if activating an ego-moving (vs. time-moving) perspective does indeed elicit higher perceived control, it may decrease perceived temporal distance to events which people are motivated to approach and increase perceived distance to events which people are motivated to avoid. Empirical support for this idea comes from a study by Ruscher [12], where participants processed either ego-moving or time-moving spatial primes and then read a vignette about a grieving mother whose young son had recently died. Participants were asked to estimate how long it would take the mother to overcome her grief and return to her daily routine. The key finding was that participants who had viewed ego-moving spatial primes (potentially therefore activating a sense of control or agency regarding an individual's movement towards the future) subsequently predicted the mother would take fewer days to move past her grief compared to participants who had viewed time-moving primes. In this case, the end of a painful period of grief and return to daily routine may represent a desirable future landmark, with mental representations of the individual actively approaching this point in time making it seem more proximal [12]. Furthermore, Boltz and Yum [13] found that participants who watched a video depicting self-movement across a stationary landscape (ego-moving condition) subsequently perceived a task deadline, an event which people may be motivated to avoid or maintain their distance from, as more temporally distant compared to participants who saw a video depicting objects moving towards a stationary observer (time-moving condition). The results of [12, 13] are consistent with the theoretical framework we have outlined, with an ego-moving (vs. time-moving) prime increasing perceived temporal distance to events which people may be motivated to approach [12], and decreasing perceived distance to events which people may be motivated to avoid [13]. However, an alternative explanation of these findings is that objective units of time may be perceived as longer when an ego-moving perspective is adopted [4]. If an objective unit of time (e.g., days or weeks) seems longer when an ego-moving perspective is adopted, a given objective temporal interval to a future event (e.g., a task

deadline [13]) may feel subjectively longer, while a given subjective temporal distance (e.g., how psychologically distant the end of a period of grief seems [12]) may be estimated to occupy fewer units of objective time.

Also of relevance to the present study, research by Xu et al. [14] (published after data collection for the present study had concluded) appeared to demonstrate that future events described using ego-moving (vs. time-moving) linguistic metaphors were perceived as more temporally distant, regardless of valence. Xu et al. argue that ego-moving descriptions increase perceived temporal distance due to a reduction in psychological arousal when individuals conceptualise themselves approaching (rather than being approached by) future events. Supporting this argument, participants perceived a difficult job interview next week as more temporally distant when it was described using an ego-moving (vs. time-moving) metaphor—mediated by decreased psychological arousal [14, Study 2]. This finding is also consistent with the theoretical framework we have outlined, with an ego-moving perspective allowing people to maintain psychological distance from potentially threatening or aversive future events. However, while we have argued that an ego-moving perspective should make positively valenced future events seem more proximal, Xu et al. [14] found that even a seemingly positive future event (delivery of a new mobile phone) was perceived as more temporally distant when framed with an ego-moving linguistic metaphor (e.g., "You are approaching the delivery day") compared to a time-moving metaphor (e.g., "The delivery day is approaching") [14, Study 5A]. However, unfortunately the authors did not test their proposed mechanism of decreased psychological arousal in this study. We suggest an alternative explanation is that the ego-moving description may have been perceived as incongruent with the future event it referred to, since a product delivery can be conceptualised as an object physically approaching the self in space (i.e., corresponding to time-moving rather than ego-moving spatial concepts). This perceived incongruency may have resulted in the ego-moving description being processed less fluently than the time-moving description, with decreased fluency increasing psychological distance of the target event [23–25]. Additionally, given the significant cross-cultural variability that has been observed in previous research examining spatial representations of time [26, 27], it is worth noting that this study [14, Study 5A], and most of the other studies reported by Xu et al. [14], used Chinese participants. It is also therefore possible that cross-cultural differences can account for this discrepancy between results of Xu et al. [14, Study 5A] and relationships between temporal perspective, perceived control, and perceived temporal distance suggested by evidence from Western samples [10–13].

## Temporal perspective and intertemporal choice

If temporal perspective influences perceived temporal distance to future events, it may have downstream effects on a range of judgements and decisions [28, 29]. In this study, we focus on the potential impact on decisions involving outcomes that unfold over different points in time–referred to as intertemporal decisions. Intertemporal decisions are common and consequential in everyday life, such as a student considering whether to go on a night out with friends or revise for an upcoming exam, or a middle-aged adult deciding whether to save money for retirement or buy a luxury car. People typically attach lower subjective value to outcomes occurring further away in time (temporal discounting), which can have significant consequences for wellbeing across the lifespan [30]. The importance of intertemporal decision-making in everyday life has inspired a wealth of behavioural and neuroimaging research, often using hypothetical monetary choice tasks in which participants decide between receiving a smaller amount of money now or a larger amount in the future (e.g., £50 today vs. £70 in 3 months) [30]. Previous research suggests it is possible to influence temporal discounting by

altering perceived temporal distance to future rewards [30–34]. For example, consistent with the idea that processing spatial information in one's environment can influence subsequent intertemporal decisions, Kim et al. [33] found that priming participants to think about short (vs. long) spatial distances in a map-based task increased preference for future rewards–mediated by a reduction in perceived temporal distance.

Based on the theoretical framework outlined in previous sections, we propose that activating an ego-moving (vs. time-moving) perspective may decrease temporal discounting by making delayed rewards feel more psychologically proximal. To our knowledge, only one study has directly investigated the association between temporal perspective and valuation of future rewards in an intertemporal choice task. In this study by Crilly [15, Study 2], participants responded to McGlone and Harding's [6] Monday-Friday question and then indicated the amount of money in one year, five years, 10 years, and 20 years they would consider equivalent to receiving $1,000 today. The results indicated that, relative to participants who gave Monday (time-moving) responses, participants who gave Friday (ego-moving) responses required significantly larger monetary values at five and 10 (though not one or 20) year intervals. These findings suggest that adopting an ego-moving temporal perspective is associated with *increased* temporal discounting–contrary to our expectations. However, as Crilly [15, Study 2] points out, temporal perspective was measured rather than manipulated, meaning the study was not able to establish causal evidence. Alternative explanations for the observed association are plausible given that individual differences in personality traits predict both temporal perspective and temporal discounting, with higher extraversion and lower conscientiousness linked to an ego-moving temporal perspective [16] and higher temporal discounting [35]. Additionally, it is possible that responding to the Monday-Friday task itself influenced subsequent intertemporal choices because participants who gave Friday responses were imagining an event being moved further into the future, while participants who gave Monday responses were imagining an event being moved closer to the present. It is possible that thinking about an event being moved further away or closer to the self in the here-and-now activates perceptions of increased or decreased psychological distance, which carries over to subsequent tasks.

## Present study

The aim of the present study was to test the hypotheses that activating an ego-moving (vs. time-moving) temporal perspective by priming corresponding spatial concepts would decrease temporal discounting and perceived temporal distance to delayed rewards in an intertemporal choice task. Data collection, hypotheses, and analysis protocols were pre-registered with the Open Science Framework (OSF) (https://osf.io/cfbms).

Hypotheses were tested using a between-groups manipulation in which participants were presented with novel map-based spatial primes designed to activate either an ego-moving or time-moving perspective (plus a control condition). These spatial primes were conceptually based on Boroditsky [5, Study 1], but aimed to reflect the type of task people may engage with on digital devices, such as following a route on a navigation app, therefore increasing the potential applicability of the findings to judgement and decision-making in everyday life. After the spatial priming task, participants responded to McGlone and Harding's [6] Monday-Friday question to examine whether the manipulation had the intended effect on temporal perspective, and then completed measures of temporal discounting, perceived wait time, and perceived control. The following confirmatory hypotheses were tested:

**H1:** The proportion of Friday responses in the Monday-Friday task will be higher in the ego-moving condition (vs. time-moving and control conditions) and lower in the time-moving condition (vs. ego-moving and control conditions).

**H2.** Perceived wait time for a future reward will feel shorter in the ego-moving condition (vs. time-moving and control conditions) and longer in the time-moving condition (vs. ego-moving and control conditions).

**H3**: Temporal discounting will be lower in the ego-moving condition (vs. time-moving and control conditions) and higher in the time-moving condition (vs. ego-moving and control conditions).

## Method

### Ethics

Ethics approval was obtained from University of Nottingham UK School of Psychology Ethics Committee (Ref: S1351). Participants provided written consent by responding to the following questions using yes/no response boxes: 1) "Have you read and understood the information sheet on the previous page?" 2) "Have you had the opportunity to ask questions about the study?" 3) "Have all your questions been answered satisfactorily (if applicable)?" 4) "Do you understand that you are free to withdraw from the study (at any time and without giving a reason)?" 5) "I give permission for my data from this study to be shared with other researchers provided my anonymity is protected." 6) "Do you agree to take part?" Participants could only progress to the study if they ticked 'yes' in response to all questions.

### Participants and statistical power

An *a priori* statistical power calculation was conducted using G*Power [36] to determine the sample size required to detect an effect of the spatial prime on temporal discounting. Based on an effect size of $d \sim 0.33$ in a study by Kim et al. [33, Study 5], which demonstrated an effect of spatial distance primes on subsequent temporal discounting, it was determined that a sample size of 573 was required to detect an effect with 95% power in a one-way ANOVA with three groups ($\alpha = .05$). To account for potential exclusions due to missing responses or failed attention checks, a target sample size of 600 participants was set. Note that this sample size also provided over 95% power to detect significant differences in perceived wait time (i.e., perceived temporal distance to a future reward), based on an effect size of $d \sim 0.39$ in Kim et al. [33, Study 5]. Based on effect sizes reported by Boroditsky [5, Study 1], the sample size also had over 95% power to detect significant differences in responses to McGlone and Harding's [6] Monday-Friday question.

UK Participants were recruited through Prolific (https://www.prolific.co/) and paid £0.88 to participate in a short online study hosted on Qualtrics (https://www.qualtrics.com/). Pre-screening criteria were specified as first language English and normal or corrected-to-normal vision (due to the requirement to identify colours in the spatial prime tasks). Data were collected from 599 participants (177 male, 416 female, 5 other, 1 not reported), with a mean age of 34.3 ($SD = 12.6$). All data were collected on 4th August 2021.

### Materials and measures

**Spatial prime manipulation.** The spatial primes consisted of a town map with locations highlighted by coloured circles and arrows. In line with Boroditsky [5, Study 1], four spatial primes were presented in each condition, with each prime displayed on a separate screen. In the ego-moving condition, participants were asked to imagine their current location was represented by a blue arrow on a map, and that they were travelling between the locations highlighted by red and orange circles. On each of the four map primes, the red and orange

circles were in separate locations with a blue arrow positioned approximately half-way between them (Fig 2). Three fill-in-the-blank style question were presented for each prime to ensure participants processed the relevant spatial information and to later exclude participants who were inattentive or failed to understand the task. The questions for each prime in the ego-moving condition were (possible response options shown in square brackets): 1) *The red circle is. . .me* [*in front of; behind*]; 2) *The orange circle is. . .me* [*in front of; behind*]; 3) *To reach the target destination, my next turn will be a. . .* [*left; right*]. So that the correct responses for question one and two were not the same for each trial, for two of the trials the blue arrow was facing the red circle, and for the other two trials the blue arrow was facing the orange circle.

In the time-moving condition, participants were asked to imagine their current location was represented by a red circle on a map, and the blue and orange arrows were people travelling towards them. On each of the four spatial primes, a red circle was displayed at a particular location on the map (e.g., school, car park), with blue and orange arrows facing the direction of travel towards the red circle (Fig 2). Again, to ensure participants processed the relevant spatial information and to exclude inattentive participants, three fill-in-the-blank questions were presented for each trial: 1) *My current location is the. . .[e.g., school; car park]; 2) The. . . arrow is closest to me [orange; blue]; 3) The blue arrow is. . . the orange arrow (in front of; behind)*. So that the correct responses for question two and three were not the same for each trial, for two of the trials the orange arrow was closer to the red circle than the blue arrow, and for the other two trials the blue arrow was closer to the red circle than the orange arrow.

In the control condition, participants were asked to look at the locations highlighted by blue, orange, and red circles (Fig 2). For each prime, two fill-in-the-blank questions were presented: 1) *The red circle is. . . of the blue circle* [*North East; North West; South East; South West*]; 2) *The orange circle is. . . of the blue circle* [*North East; North West; South East; South West*]. Compass directions were displayed in the bottom right-hand corner of the map for reference.

**Temporal perspective.** Temporal perspective was measured using McGlone and Harding's [6] Monday-Friday question. Participants were asked: "Imagine that a meeting originally scheduled for next Wednesday has been moved forward two days. What day is the meeting now that it has been rescheduled?" Monday and Friday responses were interpreted as reflecting time-moving and ego-moving perspectives, respectively. The reverse of this question (i.e., meeting moved backward two days) was also asked so that participants would think about moving a meeting both further away and closer to the present, regardless of their temporal perspective. The 'moved backward' question was included only to mitigate possible effects on perceived temporal distance caused by thinking about moving an event closer or further away from the self in the here-and-now, and was not used in any analyses.

**Intertemporal choice.** Participants were asked to imagine they had won a £50 Amazon voucher which they could either receive today, or wait three months to receive a larger amount. Participants then made 11 hypothetical choices between receiving £50 today or an amount in three months ranging from £50 to £100 in £5 increments [37]. The 11 binary choices between immediate and delayed monetary rewards were displayed vertically in ascending order of future reward value. If participants preferred £50 today to the maximum delayed value of £100, they were asked to indicate in a free-text box the amount in three months that they perceived as equivalent to receiving £50 today [37].

To quantify temporal discounting, each participant's indifference point was calculated by taking the mid-point between the values at which they switched from preferring the immediate reward to the later reward [37]. For example, if a participant preferred £50 now to £65 in three months, and £70 in three months to £50 now, an indifference point of £67.50 was assigned. The exception to this was participants who preferred £50 later to £50 now (indifference point

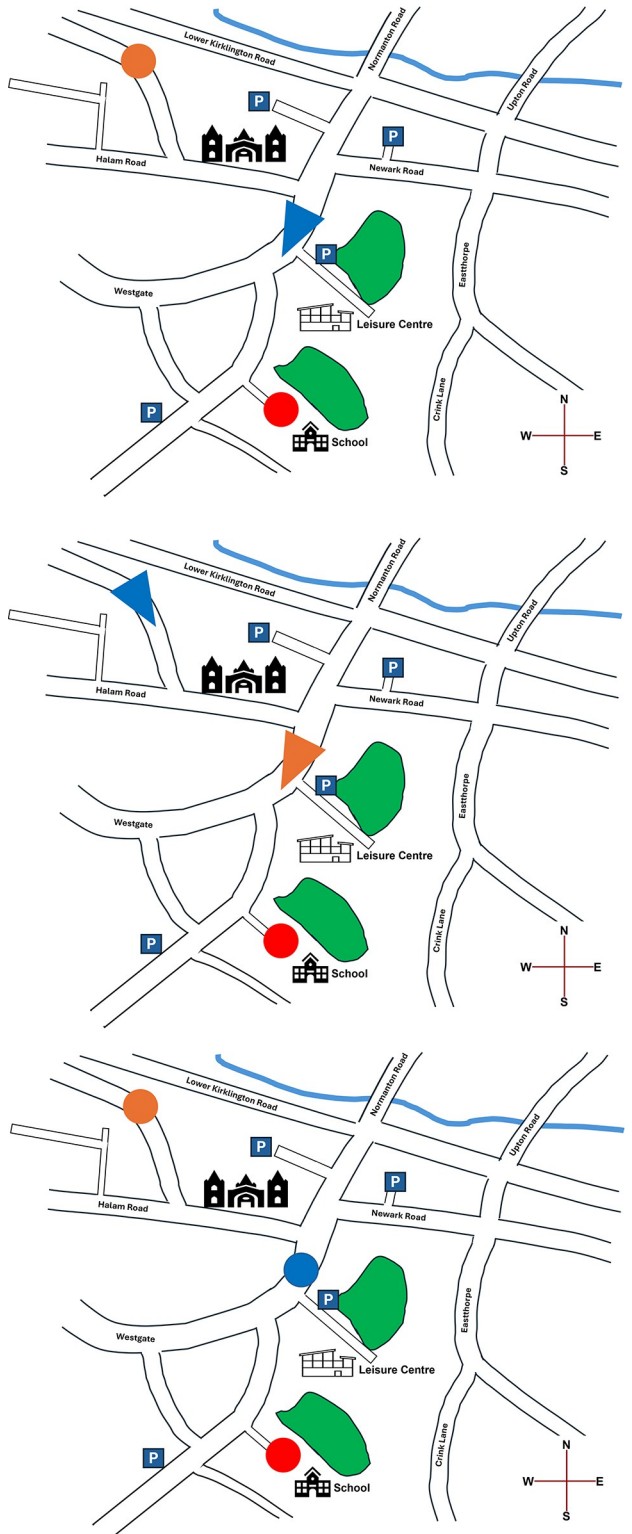

**Fig 2.** Illustrative example of a trial from ego-moving (top left), time-moving (top right), and control (bottom left) conditions. Figures were created by the study authors, loosely based on map data from Southwell town council (https://www.southwellcouncil.com/town-map/). Original spatial prime materials used in the study cannot be displayed due to copyright restrictions but are available from the corresponding author upon reasonable request.

of £50 assigned), and participants who used the free-text box to directly indicate their indifference point having rejected the delayed amount for all 11 binary choices [37]. Following Weber et al. [37], a discount factor (δ) was then calculated for each participant by applying the formula $\delta = (x_1/x_2)^{(1/(t2-t1))}$, where $x_1$ is the sooner reward amount (£50), $x_2$ is the calculated indifference point (e.g., £67.50), $t2$ is the delay for the later reward in years (0.25), and $t1$ is the delay for the sooner reward in years (0). This formula produces values between 0 and 1, where a lower discount factor indicates increased temporal discounting.

**Perceived wait time and perceived control.** Following Han and Gershoff [22], perceived wait time was measured with the question, "How long or short does a three month wait to receive your prize seem to you?" (1 = *seems very short*; 9 = *seems very long*), and perceived control over time was measured with the question, "How much control do you feel over the period of time between now and the day that you would be eligible to receive the larger prize?" (1 = *very low control*; 9 = *very high control*).

## Procedure

Participants signed up to participate in a study titled 'Map Navigation and Everyday Decisions'. The study was described as consisting of two sections. Participants were told that in the first section they would be asked to interpret some basic information on a town map, and in the second section they would be asked some questions relating to a hypothetical monetary decision. After providing consent and reporting demographic information, participants were then randomly assigned to one of three spatial prime conditions via the randomisation feature in Qualtrics. After completing the spatial primes, participants were presented with the Monday-Friday task, providing their responses in a free-text box. Next, the intertemporal choice scenario was introduced, and participants completed the perceived wait time and perceived control measures. Participants were then asked two attention check questions (correct answer in square brackets): "Please confirm what type of prize was mentioned on the previous screen [Amazon voucher]" and "Please confirm how many months you would have to wait to receive the larger prize [3 months]." Participants were then presented with the 11 binary intertemporal choices, with participants who always preferred the immediate reward subsequently presented with a free-text box to directly indicate the delayed reward amount that they considered equivalent to receiving £50 today.

## Results

### Statistical analyses

All statistical analyses were conducted in SPSS Version 28. Based on pre-registered exclusion criteria, participants were excluded from all analyses if their accuracy in response to the spatial prime questions was below 75% (n = 10). Additional pre-registered exclusion criteria for specific analyses are detailed below where applicable.

### Confirmatory hypotheses tests

**Temporal perspective.** In addition to the 10 participants excluded from all analyses, one additional participant was excluded from analysis of the Monday-Friday temporal perspective measure for responding "Wednesday", leaving a total of 588 participants.

The percentage of participants responding Monday and Friday by condition is displayed in Fig 3. A 3 x 2 Pearson's chi-square test of independence was conducted with spatial prime (control, ego-moving, time-moving) and responses to the Monday-Friday task as factors. The results revealed a significant difference in the proportion of Monday and Friday responses

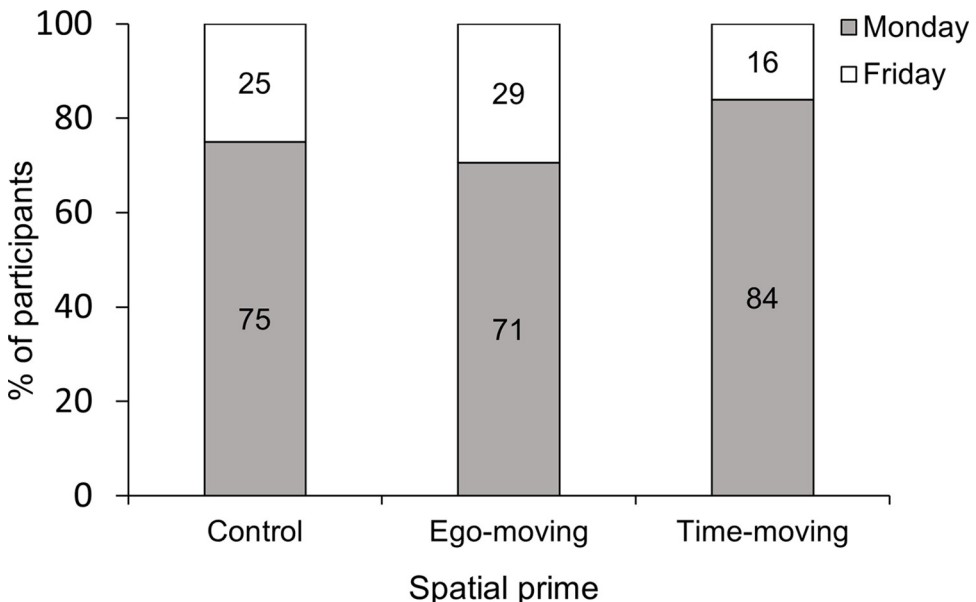

**Fig 3. Percent of Monday and Friday responses by spatial prime condition.** Monday responses indicate a time-moving perspective and Friday responses indicate an ego-moving perspective.

across conditions, $\chi^2(2) = 10.21$, $p = .006$. Partially supporting H1, follow-up chi-square tests revealed that a significantly higher percentage of participants provided Monday responses (indicating a time-moving perspective) in the time-moving condition compared to both the ego-moving condition ($\chi^2[1] = 10.06$, $p = .002$) and control condition ($\chi^2[1] = 4.78$, $p = .029$). However, contrary to expectations, there was no significant difference in the proportion of Monday and Friday responses between ego-moving and control conditions, $\chi^2(2) = 0.97$, $p = .325$.

**Perceived wait time and temporal discounting.** In addition to the 10 participants excluded from all analyses, 45 participants were excluded from analysis of perceived wait time and temporal discounting for answering one or both of the attention check questions incorrectly. Two additional participants were excluded from the perceived wait time analysis due to missing responses, and five additional participants were excluded from the temporal discounting analysis due to inconsistent responding (e.g., preferring £60 in three months to £50 now, but £50 now to £65 in three months). In a slight departure from pre-registered exclusion criteria, one participant was also excluded from temporal discounting analyses for providing a value of zero in the free-text box after rejecting all 11 delayed reward amounts, suggesting they would prefer £0 in three months to £50 today. This left a total of 542 and 538 participants for analyses of perceived wait time and temporal discounting, respectively.

To examine whether perceived wait time significantly differed between control ($M = 6.21$, $SD = 1.81$), ego-moving ($M = 6.39$, $SD = 1.69$), and time-moving ($M = 6.45$, $SD = 1.87$) conditions, a one-way ANOVA was conducted on perceived wait time with spatial prime as the between-subjects factor. The results indicated that the effect of spatial prime on perceived wait time was not significant, $F(2, 539) = 0.83$, $p = .437$, partial $\eta^2 = .003$. H2 was therefore not supported.

Next, we examined the effect of the spatial primes on discount factor, where a lower discount factor indicated increased temporal discounting. Mean discount factors by condition are displayed in Fig 4. Contrary to H3, a one-way ANOVA on discount factor, with spatial

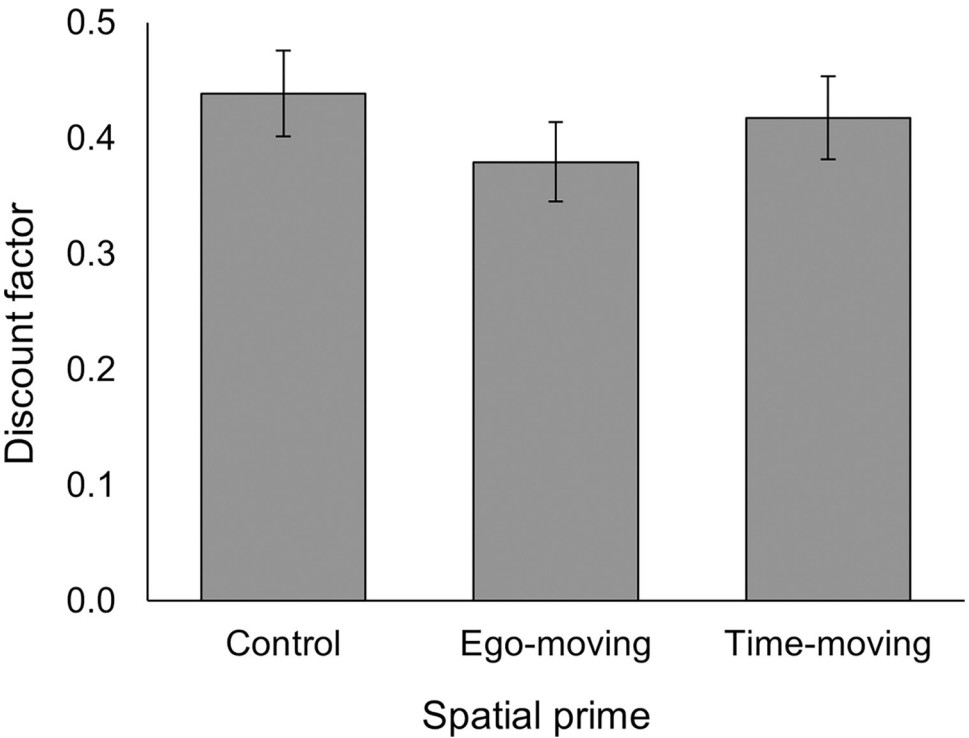

**Fig 4. Mean discount factors by spatial prime condition.** Lower discount factor scores indicate greater temporal discounting. Error bars reflect 95% confidence intervals.

prime as a between-groups factor, was not significant, $F(2, 535) = 2.68$, $p = .070$, partial $\eta^2 = .01$. Note that since this result could be considered marginally significant, we report follow-up pairwise comparisons in the supporting information (Table A in S1 Appendix) but not in the main text.

## Exploratory analysis

**Perceived control and temporal discounting.** Although there was no evidence that ego-moving or time-moving spatial primes influenced perceived wait time or temporal discounting, we proceeded to test the proposed role of perceived control in our theoretical framework. Specifically, we examined whether an ego-moving spatial prime increased perceived control over temporal distance, and whether higher perceived control was associated with lower temporal discounting via a reduction in perceived wait time.

In addition to the 10 participants excluded from all analyses, participants were excluded from analysis of perceived control if they answered either of the attention check questions incorrectly (n = 45), leaving a total of 544 participants. To examine whether perceived control significantly differed between control ($M = 4.33$, $SD = 2.45$), ego-moving ($M = 3.90$, $SD = 2.29$), and time-moving ($M = 3.99$, $SD = 2.43$) conditions, a one-way ANOVA was conducted on perceived control with spatial prime as the between-groups factor. The results indicated the effect of spatial prime on perceived control was not significant, $F(2, 541) = 1.58$, $p = .206$, partial $\eta^2 = .004$.

Next, we examined whether higher perceived control was associated with decreased temporal discounting via lower perceived wait time. Participant exclusions for these analyses were as reported in the preceding sub-section. Correlations between variables indicated that higher

**Table 1. Correlations between perceived control, perceived wait time, and temporal discounting.**

| Variables | Mean | SD | Perceived control | Perceived wait time |
|---|---|---|---|---|
| Perceived control | 4.08 | 2.39 | | |
| Perceived wait time | 6.35 | 1.80 | -.32** | |
| Discount factor | 0.41 | 0.24 | .18** | -.40** |

*SD* = standard deviation. Lower discount factor scores indicate greater temporal discounting.

** $p < .001$

perceived control was associated with lower perceived wait time, while higher perceived control and lower perceived wait time were both associated with decreased temporal discounting (Table 1).

To test whether the relationship between perceived control and temporal discounting was mediated by perceived wait time, a mediation model was computed using the PROCESS macro for SPSS (PROCESS Model 4), with percentile bootstrap confidence intervals (10,000 bootstrap samples) used to estimate 95% confidence intervals [38]. The results provided evidence for the specified indirect effect, with higher perceived control associated with lower perceived wait time, which was in turn associated with a higher discount factor (i.e., lower temporal discounting) ($b = 0.01$, $SE = 0.002$, 95% CI [0.008, 0.016]) (Fig 5). Note that this indirect effect remained significant after controlling for spatial prime condition, and spatial prime condition did not moderate any of the paths in Fig 5. Additionally, the indirect effect was not significant in an alternative model where perceived control was specified as mediating the relationship between perceived wait time and temporal discounting ($b = -0.003$, $SE = 0.002$, 95% CI [-0.007, 0.010]).

## Discussion

Against the backdrop of well-documented concerns about the replicability of many priming effects [e.g., 39, 40], this high-powered, pre-registered study lends partial support to previous findings that different temporal perspectives can be primed by prior processing of corresponding spatial information [e.g., 5, 7]. Participants in the present study who completed a map-based task which involved thinking about other people moving towards their location were

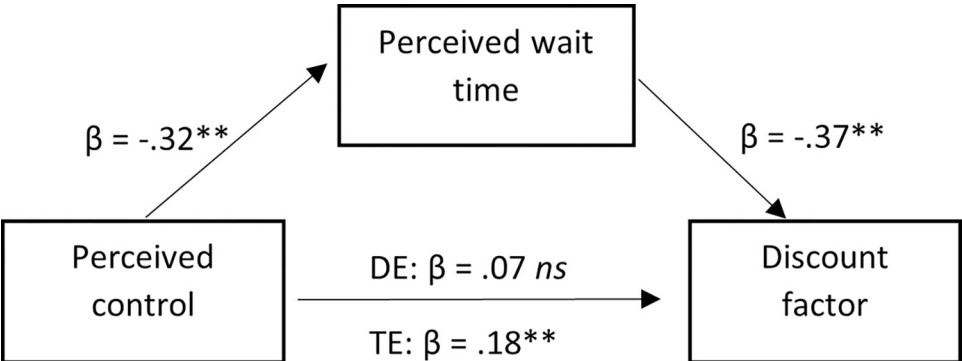

**Fig 5. Mediation model testing indirect effects of perceived control over temporal distance on discount factor, mediated by perceived wait time.** Lower discount factor scores indicate greater temporal discounting. Values are standardised coefficients reflecting the direct paths between measures. DE = direct effect, TE = total effect. *ns* = non-significant ($p > .05$). ** $p < .001$.

subsequently more likely to adopt a time-moving perspective compared to participants who thought about themselves moving between static locations, and compared to a control condition. The present study also extends previous research by using priming materials reflective of tasks which people may engage with on digital devices, thus demonstrating the potential for digital interactions to influence how people conceive of time in everyday life. For example, the time-moving prime was similar to the spatial information people may process while waiting for their car to arrive on ride-hailing app, or tracking the location of a loved one travelling towards them. However, contrary to our key hypotheses, the present study found no evidence that ego-moving or time-moving spatial primes influenced perceived wait time or discounting of future rewards in an intertemporal choice scenario. Furthermore, while the time-moving prime successfully activated the intended temporal perspective, the ego-moving prime did not. In the remainder of the discussion we outline theoretical implications of the present findings and consider potential explanations for the null effects.

## Spatial primes and temporal perspective

Although the present study found evidence that thinking about other people approaching the self primed a time-moving perspective, thinking about the self moving between static locations did not appear to prime an ego-moving perspective—contrary to previous research [5]. One possible reason for this discrepancy is that the ego-moving map navigation task in the present study required participants to process left-right spatial relationships (i.e., the direction of their next turn) in addition to front-back spatial relationships (i.e., which locations were in front/behind them on their journey). In contrast, ego-moving primes administered by Boroditsky [5, Study 1] only required participants to process front-back relationships. Since an ego-moving temporal perspective maps time onto a one-dimensional sagittal axis (i.e., past = behind, future = in front) [5], it is possible that processing lateral spatial relationships interfered with activation of the relevant spatial concepts. Furthermore, two of the ego-moving trials in the present study required participants to process left-right spatial relationships from the perspective of an arrow that was facing downwards on the screen and therefore inconsistent with their own egocentric perspective. Processing lateral spatial relationships from a conflicting perspective requires spatial perspective-taking, which is typically performed by mentally rotating the self into the to-be-adopted perspective [41]. Processing spatial relationships from a conflicting spatial perspective may have disrupted activation of an ego-moving temporal perspective, in which events are located relative to the egocentric self. It is possible that the ego-moving task used in the present study would successfully prime the intended temporal perspective if the 'direction of next turn' questions were omitted, since this would be more likely to facilitate activation of only the relevant spatial concepts.

It is also worth noting that the strong preference for Monday responses to McGlone and Harding's Monday-Friday task [6] in the present study is at odds with a meta-analysis indicating a baseline preference for Friday responses among English speakers [17]. It is possible that this discrepancy is due to differences in demographic characteristics, with many previous studies using undergraduate samples [e.g., 5–7, 9, 18], compared to the present study where the mean sample age was 34 and over 40% of participants were in full-time employment (see Table B in S1 Appendix). These differences in demographic characteristics are potentially relevant to the discrepancy in temporal perspectives across studies because previous research has found that people in full-time employment are more likely to adopt a time-moving perspective [16], with exploratory analysis of the present data also indicating that increased age was associated with greater odds of providing a time-moving (i.e., Monday) response (see Table C in S1 Appendix).

### Temporal perspective and intertemporal choice

The apparent failure of the ego-moving prime to activate the target temporal perspective may explain why the experimental manipulation did not produce the hypothesised effects on temporal discounting or perceived wait time. However, regardless of the effectiveness of the spatial primes, if perceived wait time and temporal discounting were related to participants' current temporal perspective, scores on these measures should have been predicted by their responses (i.e., Monday vs. Friday) to McGlone and Harding's [6] ambiguous scheduling question, which was not the case (see Table D in S1 Appendix). The absence of a significant association between measured temporal perspective and temporal discounting is contrary to Crilly's [15, Study 2] observation that Friday (vs. Monday) responses predicted decreased valuation of future monetary rewards in an intertemporal choice task. One possible explanation for the discrepancy between results is that Crilly [15, Study 2] used longer delay lengths (1, 5, 10, 20 years) than in the present study (3 months), with significant associations observed at five and ten-year intervals. It may be interesting for future research to examine whether the relationship between temporal perspective and intertemporal choice is moderated by delay length.

The absence of spatial priming effects on perceived wait time is somewhat inconsistent with previous evidence that processing ego-moving and time-moving spatial primes influenced subsequent temporal distance judgements [12, 13]. One potentially interesting difference between studies is that the future outcome in both [12, 13]–but not the present study–related to actions that could be influenced by the agent in the specified scenario. Specifically, in [12], participants read a vignette about a bereaved mother who would have to overcome her grief to return to her daily routine, while in [13], participants were required to complete a task before a deadline. Conversely, in the present study, receipt of the delayed reward was unrelated to active engagement of the self during the intervening passage of time. It is possible that an ego-moving perspective elicits greater perceived control over one's progression towards a particular goal or outcome, rather than an illusory perception of control over their movement through time *per se*. If this is the case, an ego-moving perspective may have a greater effect on perceived temporal distance to future outcomes that are connected to actions that can be controlled by the self. For example, a student who conceptualises time from an ego-moving perspective may expect to finish an assignment more quickly, which may in turn make the deadline feel subjectively further away, since the temporal interval to the deadline seems long relative to the time the task is expected to take. Indeed, this is consistent with Boltz and Yum's [13] finding that in addition to perceiving a task deadline as more psychologically distant, participants primed with an ego-moving perspective estimated they would complete the task more quickly. Conversely, extending the aforementioned example to when the student has now submitted their assignment, adopting an ego-moving (vs. time-moving) perspective may have limited impact on perceived temporal distance to the date results are released, since this future event is unrelated to actions that can be influenced by the self during the temporal interval.

### Perceived control and intertemporal choice

Exploratory analysis indicated that greater perceived control was associated with decreased temporal discounting, with the relationship mediated by lower perceived temporal distance to future rewards. While previous research has found inconsistent evidence for a relationship between perceived control and temporal discounting [e.g., 15, 42, 43], participants in the present study were asked more specifically about perceived control over the period of time between now and the day a future reward would be received—with this particular measure of perceived control associated with perceived temporal distance in previous research [22]. Higher

perceived control may influence perceived temporal distance based on an overgeneralisation from the spatial domain, where a high degree of control enables people to decrease their physical distance to things they are motivated to approach and maintain or increase their physical distance from things they are motivated to avoid [22]. It is also possible that individuals differ in the extent to which they feel their subjective experience of time is malleable. For example, someone may experience a high sense of control over temporal distance if they feel they can make a period of time feel subjectively shorter (e.g., by occupying themselves with other interesting activities) or longer (e.g., by being mindful in the present moment). If this is the case, it may be possible to influence temporal discounting by manipulating perceived control over one's subjective experience of time. Future research could develop more detailed scale measures of perceived control over subjective time to examine correlations with psychological distance of future events and temporal discounting, in order to clarify the nature of these relationships. Additionally, experimental research could perhaps manipulate perceived control over subjective time by having participants write about a past experience in which they remember having control over time, e.g., had successfully "killed time" (vs. a control condition), before then completing an intertemporal choice task.

## Limitations

A limitation of the present study is that temporal perspective was measured with a single item–McGlone and Harding's [6] Monday-Friday question. While many previous studies have relied on this measure of temporal perspective and assumed that Monday responses reflect a time-moving perspective, Núñez et al. [18] argue that Monday responses may also reflect a non-egocentric perspective in which events are mentally represented according to their order in a sequence, without reference to the self in the present moment (e.g., "Christmas comes before New Year's"). In this non-egocentric perspective, moving an event forward means moving it closer to the front of the sequence, therefore also resulting in Wednesday's meeting being moved to Monday [18]. As the time-moving prime in the present study involved processing information about the order of objects (i.e., arrows) relative to each other, it is possible that the increased rate of Monday responses in this condition reflected activation of a non-egocentric perspective rather than a time-moving perspective. Since spatial schemas underpinning this non-egocentric temporal perspective relate to the position of objects relative to each other rather than relative to the self [18], they may not be relevant to egocentric temporal distance judgements (e.g., when judging how far away receipt of a future reward feels).

Another potential limitation of the present research is that we used a temporal discounting task involving a single delay length (3 months). We chose this temporal discounting task because we believed the preceding spatial primes may have a larger effect on perceived temporal distance and temporal discounting when participants imagined a single specific scenario rather than switching between multiple delay lengths, as is the case in other common intertemporal choice tasks [e.g., 44, 45]. Furthermore, we thought it was important that the temporal discounting task was relatively brief since any spatial priming effects may only be short-lived. However, the disadvantage of a single delay length is that it would not have been possible to detect spatial priming effects occurring at shorter (e.g., 1 week) or longer (e.g., 5 years) delays.

## Conclusion

The results of this high-powered, pre-registered study partly support previous research by demonstrating that processing spatial information influences how people mentally represent the passage of time, with participants who imagined other people moving towards their location on a map subsequently more likely to conceptualise future temporal events as

approaching the self. However, in contrast to previous research, there was no evidence that priming spatial concepts relating to self-movement between static locations led people to mentally represent themselves as approaching future events. Additionally, contrary to our other pre-registered hypotheses, the spatial primes had no effect on either perceived temporal distance or temporal discounting of delayed rewards in a subsequent intertemporal choice task. Exploratory analysis indicated that greater perceived control over the period of time between the present and receipt of a future reward was associated with lower temporal discounting, mediated by lower perceived wait time, suggesting a possible area for future research into individual differences and interventions in intertemporal decision-making.

## Supporting information

**S1 Appendix. Supplementary analyses.**
(DOCX)

## Author Contributions

**Conceptualization:** Daniel Fletcher, Robert Houghton, Alexa Spence.

**Formal analysis:** Daniel Fletcher.

**Investigation:** Daniel Fletcher.

**Methodology:** Daniel Fletcher.

**Supervision:** Robert Houghton, Alexa Spence.

**Writing – original draft:** Daniel Fletcher.

**Writing – review & editing:** Daniel Fletcher, Robert Houghton, Alexa Spence.

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
