## [Decision Letter · Decision Letter 0]

27 Feb 2024

PONE-D-23-41597Approaching future rewards or waiting for them to arrive: Spatial representations of time and intertemporal choicePLOS ONE

Dear Dr. Fletcher,

Thank you for submitting your manuscript to PLOS ONE. After careful consideration, we feel that it has merit but does not fully meet PLOS ONE’s publication criteria as it currently stands. Therefore, we invite you to submit a revised version of the manuscript that addresses the points raised during the review process.

We look forward to receiving your revised manuscript.

Kind regards,

Yan Wang

Academic Editor

PLOS ONE

2. We note that Figure 2 (bottom left), 2 (top left), 2 (top right), and S2 Stimuli in your submission contain [map/satellite] images which may be copyrighted. All PLOS content is published under the Creative Commons Attribution License (CC BY 4.0), which means that the manuscript, images, and Supporting Information files will be freely available online, and any third party is permitted to access, download, copy, distribute, and use these materials in any way, even commercially, with proper attribution. For these reasons, we cannot publish previously copyrighted maps or satellite images created using proprietary data, such as Google software (Google Maps, Street View, and Earth). For more information, see our copyright guidelines: http://journals.plos.org/plosone/s/licenses-and-copyright.

a. You may seek permission from the original copyright holder of Figure 2 (bottom left), 2 (top left), 2 (top right), and S2 Stimuli  to publish the content specifically under the CC BY 4.0 license.  

Reviewers' comments:

Reviewer's Responses to Questions

**Comments to the Author**

1. Is the manuscript technically sound, and do the data support the conclusions?

Reviewer #1: Partly

Reviewer #2: Yes

2. Has the statistical analysis been performed appropriately and rigorously? 

Reviewer #1: Yes

Reviewer #2: Yes

3. Have the authors made all data underlying the findings in their manuscript fully available?

Reviewer #1: Yes

Reviewer #2: Yes

4. Is the manuscript presented in an intelligible fashion and written in standard English?

Reviewer #1: Yes

Reviewer #2: Yes

5. Review Comments to the Author

Reviewer #1: Comments are as follows:

1. The paper needs to be streamlined and highlight your work. The current description is too redundant.

2. Need more references for work in the past 3 years.

3. What is the contribution of this paper? need to be emphasized.

4. It is best to put pictures together with corresponding content to more clearly demonstrate the work of the paper.

Reviewer #2: This manuscript presents a study investigating how spatial primes (ego-moving vs. time-moving perspectives) influence temporal discounting, perceived wait time, and perceived control in decision-making. Here are my observations and suggestions for improvement across various aspects of the manuscript:

Clarity and Structure

The manuscript is well-structured, with clear sections delineating the introduction, methods, results, and discussion. The logical flow from the introduction of spatial metaphors and temporal perspectives to the hypotheses and methodology is commendable. However, the discussion section could benefit from a more detailed comparison with previous studies, especially in areas where your findings diverge.

Argumentation

Your argumentation is strong, especially in linking spatial primes with temporal perspectives and their potential effects on decision-making. The literature review effectively sets the stage for your hypotheses. Nonetheless, the discussion could further explore the implications of the findings, particularly the lack of significant effects of the spatial primes on temporal discounting and perceived wait time. Discussing alternative explanations and potential moderating variables could enrich this section.

Methodology

The methodology is robust, with a clear description of the participant selection, spatial prime manipulation, and measurement of temporal perspectives and discounting. The pre-registration and power analysis enhance the credibility of the study. However, consider discussing the limitations of the spatial prime manipulation, especially regarding the ego-moving condition's effectiveness, and how it might be improved in future studies.

Key Findings

It's crucial to emphasize the novelty of your findings in the discussion, especially the absence of significant effects from spatial primes on temporal discounting and perceived wait time. This contrasts with some previous literature and invites further investigation into the conditions under which spatial primes might affect temporal perspectives and decision-making.

Future Directions

The manuscript hints at future research avenues, such as interventions to increase perceived control over time to reduce temporal discounting. Expanding on these suggestions by proposing specific experimental designs or theoretical models could make this section more impactful.

Technical Corrections

Ensure consistency in referencing figures, tables, and supplementary materials. Double-check that all mentioned resources are correctly cited and accessible.

Consider shortening some of the more detailed explanations in the results section to improve readability, summarizing key points more succinctly.

Overall Impression

The manuscript contributes valuable insights to the literature on temporal perspectives and decision-making, despite some findings being contrary to initial hypotheses. Its strengths lie in the rigorous methodology and the thoughtful consideration of alternative explanations for the observed results. Addressing the suggested improvements could enhance the manuscript's impact and provide clearer directions for future research in this intriguing area of study.

6. PLOS authors have the option to publish the peer review history of their article (what does this mean?). If published, this will include your full peer review and any attached files.

Reviewer #1: No

Reviewer #2: **Yes: **Wonsok Frank Jee

---

## [Author Response · Author response to Decision Letter 0]

19 Mar 2024

Dear Dr Wang,

We would like to thank you for the opportunity to submit a revised version of our manuscript, and the reviewers for their helpful and constructive comments. We believe that addressing these comments has resulted in a significantly improved manuscript. 

The primary changes to the manuscript were to enhance the discussion section by including more detailed comparisons of the findings with prior research and to discuss possible reasons for the null experimental effects. We have also made text changes throughout the manuscript to enhance clarity, and supplied replacement figures to be used for illustrative purposes where potential copyright issues were raised.

Responses to each editor and reviewer comment are provided below. 

We look forward to hearing from you in due course.

Yours sincerely,

Daniel Fletcher (Corresponding Author)

Editor: Journal Requirements

Data Availability:

Author Comment:

The dataset is now available from UK Data Service data repository using the link below:

https://reshare.ukdataservice.ac.uk/856861/

Author Comment:

We have updated the title page to match the PLOS ONE style template. 

2. We note that Figure 2 (bottom left), 2 (top left), 2 (top right), and S2 Stimuli in your submission contain [map/satellite] images which may be copyrighted. All PLOS content is published under the Creative Commons Attribution License (CC BY 4.0), which means that the manuscript, images, and Supporting Information files will be freely available online, and any third party is permitted to access, download, copy, distribute, and use these materials in any way, even commercially, with proper attribution. For these reasons, we cannot publish previously copyrighted maps or satellite images created using proprietary data, such as Google software (Google Maps, Street View, and Earth). For more information, see our copyright guidelines:http://journals.plos.org/plosone/s/licenses-and-copyright.

Author Comment:

We have now supplied replacement images for Figure 2 (top left, top right, bottom left) for illustrative purposes and removed S2 Stimuli from the submission. We have updated the Fig 2 caption to read:

“Fig 2. Illustrative example of a trial from ego-moving (top left), time-moving (top right), and control (bottom left) conditions. Original spatial prime materials used in the study cannot be displayed due to copyright restrictions but are available from the corresponding author upon reasonable request.”

3. Please include captions for your Supporting Information files at the end of your manuscript, and update any in-text citations to match accordingly. 

Author Comment:

We have added the following information caption added to the manuscript:

“S1 Appendix. Supplementary analyses”

In-text citations refer to S1 Appendix and a specific table within this document where applicable.

Author Comment:

We have made formatting changes to the references where required and added DOIs to aid accessibility . We have added 3 references in response to Reviewer 1’s second comment and an additional 5 references to strengthen our arguments. We have also changed the order of some references to match changes to the order of in-text citations. The reference list remains otherwise unchanged.

The added references are as follows:

Pfaltz MC, Plichta MM, Bockisch CJ, Jellestad L, Schnyder U, Stocker K. Processing of an ambiguous time phrase in posttraumatic stress disorder: Eye movements suggest a passive, oncoming perception of the future. Psychiatry Research. 2021;299: 113845. https://doi.org/10.1016/j.psychres.2021.113845

Maglio SJ. Psychological distance in consumer psychology: Consequences and antecedents. Consumer Psychology Review. 2020;3(1): 108–125. https://doi.org/10.1002/arcp.1057

Baron S, Everett BC, Latham AJ, Miller K, Tierney H, Veng J. Moving ego versus moving time: investigating the shared source of future-bias and near-bias. Synthese. 2023;202. https://doi.org/10.1007/s11229-023-04286-0

Lee A, Ji L-J. Moving away from a bad past and toward a good future: Feelings influence the metaphorical understanding of time. Journal of Experimental Psychology: General. 2014;143: 21–26. doi:https://doi.org/10.1037/a0032233

Lempert KM, Phelps EA. The Malleability of Intertemporal Choice. Trends in Cognitive Sciences. 2016;20(1): 64–74. https://doi.org/10.1016/j.tics.2015.09.005

Gu Y, Zheng Y, Swerts M. Which Is in Front of Chinese People, Past or Future? The Effect of Language and Culture on Temporal Gestures and Spatial Conceptions of Time. Cognitive Science. 2019;43(12). https://doi.org/10.1111/cogs.12804

Boroditsky L, Fuhrman O, McCormick K. Do English and Mandarin speakers think about time differently? Cognition. 2011;118(1): 123–129. https://doi.org/10.1016/j.cognition.2010.09.010

Du W, Green L, Myerson J. Cross-Cultural Comparisons of Discounting Delayed and Probabilistic Rewards. The Psychological Record. 2002;52: 479–492. https://doi.org/10.1007/bf03395199

Reviewer 1:

1. The paper needs to be streamlined and highlight your work. The current description is too redundant.

Author Comment:

We thank the reviewer for pointing out the need to streamline the paper by removing redundancy. We have reduced text in the Method and Results sections where possible to do so without removing detail which may be important to readers. We have also made changes to the text throughout the manuscript which we hope enhance clarity. 

2. Need more references for work in the past 3 years.

Author Comment:

We have now added the following references from 2020 onwards:

Pfaltz MC, Plichta MM, Bockisch CJ, Jellestad L, Schnyder U, Stocker K. Processing of an ambiguous time phrase in posttraumatic stress disorder: Eye movements suggest a passive, oncoming perception of the future. Psychiatry Research. 2021;299: 113845. https://doi.org/10.1016/j.psychres.2021.113845

Maglio SJ. Psychological distance in consumer psychology: Consequences and antecedents. Consumer Psychology Review. 2020;3(1): 108–125. https://doi.org/10.1002/arcp.1057

Baron S, Everett BC, Latham AJ, Miller K, Tierney H, Veng J. Moving ego versus moving time: investigating the shared source of future-bias and near-bias. Synthese. 2023;202. https://doi.org/10.1007/s11229-023-04286-0

3. What is the contribution of this paper? need to be emphasized.

Author Comment:

We thank the reviewer for highlighting the need to make the contribution of the manuscript clearer. We have now updated the discussion to more clearly situate the present findings within the wider literature – particularly highlighting where the present results differ from previous research and discussing possible theoretical implications of these differences. We refer the reviewer to the Discussion section of the revised manuscript. 

4. It is best to put pictures together with corresponding content to more clearly demonstrate the work of the paper.

Author Comment:

We thank the reviewer for this comment and agree this would enhance clarity in the final article. We were required to provide figures separately to the main manuscript for initial submission to PLOS ONE, though in the final article figures would appear above their corresponding caption.

Reviewer 2: 

Clarity and Structure

The manuscript is well-structured, with clear sections delineating the introduction, methods, results, and discussion. The logical flow from the introduction of spatial metaphors and temporal perspectives to the hypotheses and methodology is commendable. However, the discussion section could benefit from a more detailed comparison with previous studies, especially in areas where your findings diverge.

Author Comment:

We thank the reviewer for these comments and the helpful suggestion to improve the discussion by including a more detailed comparison with prior research. We have updated the manuscript accordingly to include a detailed comparison between the present findings and previous research – highlighting discrepancies in results and considering possible explanations for these differences. We refer the reviewer to the Discussion section of the revised manuscript.

Argumentation

Your argumentation is strong, especially in linking spatial primes with temporal perspectives and their potential effects on decision-making. The literature review effectively sets the stage for your hypotheses. Nonetheless, the discussion could further explore the implications of the findings, particularly the lack of significant effects of the spatial primes on temporal discounting and perceived wait time. Discussing alternative explanations and potential moderating variables could enrich this section.

Author Comment:

We have updated the manuscript to explore the lack of significant effects on perceived wait time and temporal discounting in more detail. In particular, we consider possible reasons for the absence of significant effects, draw comparisons with previous literature, and discuss some possible moderators. We refer the reviewer to the sub-section ‘Temporal perspective and intertemporal choice in the Discussion section of the revised manuscript. 

Methodology

The methodology is robust, with a clear description of the participant selection, spatial prime manipulation, and measurement of temporal perspectives and discounting. The pre-registration and power analysis enhance the credibility of the study. However, consider discussing the limitations of the spatial prime manipulation, especially regarding the ego-moving condition's effectiveness, and how it might be improved in future studies.

Author Comment:

We have added the following text to the discussion which outlines possible reasons the ego-moving prime was not effective, in contrast to previous research:

“One possible reason for this discrepancy is that the ego-moving map navigation task in the present study required participants to process left-right spatial relationships (i.e., the direction of their next turn) in addition to front-back spatial relationships (i.e., which locations were in front/behind them on their journey). In contrast, ego-moving primes administered by Boroditsky [5, Study 1] only required participants to process front-back relationships. Since an ego-moving temporal perspective maps time onto a one-dimensional sagittal axis (i.e., past = behind, future = in front) [5], it is possible that processing lateral spatial relationships interfered with activation of the relevant spatial concepts. Furthermore, two of the ego-moving trials in the present study required participants to process left-right spatial relationships from the perspective of an arrow that was facing downwards on the screen and therefore inconsistent with their own egocentric perspective. Processing lateral spatial relationships from a conflicting perspective requires spatial perspective-taking, which is typically performed by mentally rotating the self into the to-be-adopted perspective [41]. Processing spatial relationships from a conflicting spatial perspective may have disrupted activation of an ego-moving temporal perspective, in which events are located relative to the egocentric self. It is possible that the ego-moving task used in the present study would successfully prime the intended temporal perspective if the ‘direction of next turn’ questions were omitted, since this would be more likely to facilitate activation of only the relevant spatial concepts.”

Key Findings

It's crucial to emphasize the novelty of your findings in the discussion, especially the absence of significant effects from spatial primes on temporal discounting and perceived wait time. This contrasts with some previous literature and invites further investigation into the conditions under which spatial primes might affect temporal perspectives and decision-making.

Author Comment:

We have updated the manuscript to highlight where the present findings differ from existing literature, and what may explain these discrepancies. We refer the reviewer to the subsection ‘Temporal perspective and intertemporal choice in the Discussion section of the revised manuscript.

Future Directions

The manuscript hints at future research avenues, such as interventions to increase perceived control over time to reduce temporal discounting. Expanding on these suggestions by proposing specific experimental designs or theoretical models could make this section more impactful.

Author Comment:

We thank the reviewer for this helpful suggestion. We have now added the following text to suggest a more specific direction for further research:

“Future research could develop more detailed scale measures of perceived control over subjective time to examine correlations with psychological distance of future events and temporal discounting, in order to clarify the nature of these relationships. Additionally, experimental research could perhaps manipulate perceived control over subjective time by having participants write about a past experience in which they remember having control over time, e.g., had successfully “killed time” (vs. a control condition), before then completing an intertemporal choice task.”

Technical Corrections

Ensure consistency in referencing figures, tables, and supplementary materials. Double-check that all mentioned resources are correctly cited and accessible.

Consider shortening some of the more detailed explanations in the results section to improve readability, summarizing key points more succinctly.

Author Comment:

We have reformatted references where required and added DOIs to aid accessibility. We have also slightly reduced text in the Results and Methods section where possible to do so without removing detail which may be important to readers. 

---

## [Editor Report · Decision Letter 1]

22 Mar 2024

Approaching future rewards or waiting for them to arrive: Spatial representations of time and intertemporal choice

PONE-D-23-41597R1

Dear Dr. Fletcher,

We’re pleased to inform you that your manuscript has been judged scientifically suitable for publication and will be formally accepted for publication once it meets all outstanding technical requirements.

Kind regards,

Yan Wang

Academic Editor

PLOS ONE

---

## [Editor Report · Acceptance letter]

27 Mar 2024

PONE-D-23-41597R1 

PLOS ONE

Dear Dr. Fletcher, 

I'm pleased to inform you that your manuscript has been deemed suitable for publication in PLOS ONE. Congratulations! Your manuscript is now being handed over to our production team.

Kind regards, 

on behalf of

Dr. Yan Wang 

Academic Editor

PLOS ONE